# Protective Effects of *Bauhinia forficata* on Bone Biomechanics in a Type 2 Diabetes Model

**DOI:** 10.3390/ph18111724

**Published:** 2025-11-13

**Authors:** Isadora Castaldi Sousa, Letícia Pitol-Palin, Fábio Roberto de Souza Batista, Odir Nunes de Oliveira Filho, Sabrina Cruz Tfaile Frasnelli, Victor Eduardo de Souza Batista, Dóris Hissako Matsushita, Roberta Okamoto

**Affiliations:** 1Department of Basic Sciences, Araçatuba Dental School, São Paulo State University, Araçatuba 16066-840, Brazil; isadora.c.sousa@unesp.br (I.C.S.); fabiorsbatista@gmail.com (F.R.d.S.B.); satfaile@yahoo.com.br (S.C.T.F.);; 2Department of Diagnosis and Surgery, Araçatuba Dental School, São Paulo State University, Araçatuba 16015-050, Brazil; odir.nunes@unesp.br; 3Department of Prosthodontics, Presidente Prudente Dental School, University of Western São Paulo, Presidente Prudente 19050-680, Brazil; victor_edsb@hotmail.com

**Keywords:** diabetes mellitus, Type 2, phytotherapeutic, natural products, bone remodeling, bone matrix, dental implants, animal model

## Abstract

The use of herbal medicines has gained popularity, both in science and among the public, as a natural alternative for the treatment of numerous diseases, including type 2 diabetes. **Objective**: The objective of this study was to evaluate peri-implant and long bone biomechanics in type 2 diabetic animals, treated or not with *Bauhinia forficata*. **Methods**: Thirty-two rats were allocated into four groups: normoglycemic (NG), normoglycemic + *Bauhinia forficata* (NGBf), type 2 diabetes (T2D), and T2D + *Bauhinia forficata* (T2DBf). Diabetes was induced using a cafeteria diet and streptozotocin (35 mg/kg). *Bauhinia forficata* tea (50 g/L) was administered to the NGBf and T2DBf groups. After 14 days, titanium implants were installed in the tibial metaphysis of all animals. Biomechanical analysis (removal torque), computerized microtomography, three-point bending test, confocal microscopy, and real-time PCR were performed. The results were tabulated, and a statistical test was conducted with a significance level of 5%. **Results**: *Bauhinia forficata* significantly improved the weight and blood glucose levels of the animals. In terms of biomechanics and the microarchitecture of long bones, T2D did not impair bone metabolism, and the use of the therapy did not cause significant changes in the parameters evaluated. However, T2D promoted significant impairment in the structural, biomechanical, and molecular characteristics of the peri-implant repair process, and the use of *Bauhinia forficata* increased the parameters evaluated in T2DBf. **Conclusions**: Type 2 diabetes mellitus significantly compromises peri-implant bone repair, with no influence on the metabolism of long bones, and *Bauhinia forficata* acts positively on both the etiopathogenesis of the disease and the tissue response to bone repair.

## 1. Introduction

Diabetes mellitus is a group of diseases characterized by high glucose levels in the blood, known as hyperglycemia, which happens due to problems with insulin production and action. During type 2 diabetes, the pancreas initially tries to compensate for insulin resistance, but factors such as obesity and sedentary lifestyles lead to failures among beta cells [1] and decreased insulin production, resulting in high glucose levels [2]. The increase is due to sedentary lifestyles, obesity, population aging processes, and the longer lifespan of diabetic patients. Complications include microvascular issues (retinopathy, nephropathy, neuropathy) and macrovascular problems (myocardial infarction, stroke, peripheral artery disease). The primary cause of death among diabetics is atherosclerotic cardiovascular disease [3].

This metabolic condition has a negative impact on bone tissue health. Although the risk of fracture is not increased in individuals with type 2 diabetes, it acts on tissue metabolic pathways [4,5,6,7], even though bone mass is normal, and causes a significant impact on bone homeostasis, affecting various mechanisms that are essential for bone tissue. Collagen properties are compromised due to the accumulation of advanced glycation products (AGEs), the formation of irreversible cross-links, and reactive oxygen species (ROS) [5]. These factors together weaken bone matrix deposition by interfering with osteoblast function and producing inflammatory cytokines and ROS that are related to bone resorption [8,9].

In the dental context, type 2 diabetes has a significant impact on oral health. Diabetic patients are more likely to develop periodontitis and peri-implantitis, conditions that lead to bone loss around teeth and implants [10,11]. This condition also complicates the osseointegration process: in procedures such as the installation of dental implants, the exacerbation of the inflammatory phase leads to delayed tissue repair, since collagen fibers will not be produced correctly, leading to a mineralization process that is also deficient [11]. Hyperglycemia, associated with type 2 diabetes, can also compromise the body’s immune response, thus increasing the risk of infections and complicating dental treatments [11,12].

Current therapeutic approaches for type 2 diabetes aim to improve glycemic control, enhance insulin sensitivity, and prevent long-term complications [1,2,3]. Conventional management includes lifestyle modifications, such as a balanced diet and regular physical activity, that can be combined with pharmacological agents, including metformin, sulfonylureas, thiazolidinediones, DPP-4 inhibitors, GLP-1 receptor agonists, and SGLT2 inhibitors [6,10]. Although these treatments are effective, they can present adverse effects, limited long-term efficacy, and high costs, which may compromise patient adherence.

Notably, this indicates that phytotherapy has emerged as a complementary or alternative strategy for T2D management. Several medicinal plants and natural compounds have demonstrated hypoglycemic, antioxidant, and anti-inflammatory properties, acting through multiple molecular pathways to improve glucose metabolism and reduce diabetic complications [13]. Such characteristics make herbal therapies a promising and accessible adjunct to conventional pharmacological treatment. This is due to the potential of these products to minimize side effects and reactions in the patient [13].

*Bauhinia forficata* Link (Bf) is a native Brazilian plant found in the Cerrado region that is recommended by SUS (Brazilian Unified Health System) as part of the treatment for type 2 diabetes. Its bioactive compounds, such as lactones, terpenoids, glycolipids, and, in particular, flavonoids, are attributed to these beneficial effects (hypoglycemic, antidiabetic, antioxidant, and anti-inflammatory) [14,15,16,17,18,19,20,21,22,23,24,25]. Beyond glycemic control, *Bauhinia forficata* offers a promising multimodal approach capable of modulating oxidative stress, inflammatory pathways, and tissue repair mechanisms, factors closely linked to diabetic complications and impaired bone metabolism [14]. Moreover, their accessibility, low cost, and cultural acceptance in traditional medicine strengthen their relevance as complementary strategies to conventional pharmacological treatments.

Natural compounds can modulate osteoblastic differentiation, reduce oxidative stress, and control inflammation, all of which are key factors in fracture healing, particularly under diabetic conditions. *Bauhinia forficata* contains kaempferol, a flavonoid with antidiabetic effects, which fights free radicals and modulates important cell signaling pathways, such as nuclear factor Kappa-B (NF-kB), related to OPG/RANK-L pathway activation [14,25,26]. This suggests a potential role for *Bauhinia forficata* in bone tissue. Different compounds have been proven in bone tissue modulation, such as genistein [27], curcumin [28], carotenoid lycopene [29], yerba mate [30], and countless other species. The osteoprotective effects of herbal compounds vary through the modulation of osteogenic markers and antioxidant mechanisms, interfering with various pathways of bone metabolism [27,28,29,30]. These findings reinforce the therapeutic potential of phytochemicals as complementary strategies to optimize bone repair and mitigate the deleterious effects of systemic diseases such as type 2 diabetes.

In this context, considering the benefits of *Bauhinia forficata* therapy and the lack of studies proving its effectiveness in type 2 diabetes models and bone metabolism, the necessity for the present study is evident. Thus, the aim of this study was to evaluate peri-implant and long bone biomechanics in type 2 diabetic animals, treated or not with *Bauhinia forficata*.

## 2. Results

### 2.1. Body Weight

The NG and NGBf groups showed a gradual increase in their weight, as expected. T2D and T2DBf registered lower weight averages than the normoglycemic groups. The following differences were observed for the NG group: Day 0 vs. Day 70 (*p* = 0.0025); Day 28 vs. Day 70 (*p* = 0.0113); Day 42 vs. Day 70 (*p* = 0.037). In the NGBf group, a statistically significant difference was observed between Day 0 and Day 42 (*p* = 0.0023). In the T2D and T2DBf groups, there were no statistically significant differences between the animals’ weights during the experimental period (Figure 1 and Table 1).

### 2.2. Glycemic Level

In the NG group, there was no statistically significant difference (*p* > 0.05) in the mean glycemic values on Day 0 compared to Day 70. In the NGBf group, there was a small increase in glycemic values, with a statistically significant difference (*p* = 0.0255) at the end of the experiment; however, the animals still presented a normoglycemic state. The T2D group showed an increase in blood glucose after 3 weeks on the cafeteria diet, with no statistical difference, and with the application of streptozotocin (STZ), there was an increase in the group’s glycemic values, confirming the systemic picture of T2D. This increase in the glycemic index continued until euthanasia, where the T2D animals had values above 400 mg/dL (*p* < 0.0001). The phytotherapeutic treatment in T2DBf, on the other hand, promoted a decrease in glycemic levels (185.1 mg/dL) at the end of the experiment compared to the values prior to the induction of T2D, presenting a small statistically significant difference (*p* = 0.0287) (Figure 2 and Table 2).

### 2.3. Long Bone Biomechanical Analyses

#### 2.3.1. Three-Point Bending Test

*Maximum Breaking Force (N)*: The maximum breaking force values were obtained in the femoral shaft region, reflecting the maximum load that the bone can withstand before suffering a complete rupture. The analysis showed a statistically significant difference when comparing NG vs. T2DBf (*p* = 0.185), NGBf vs. T2D (*p* = 0.136), and NGBf vs. T2DBf (*p* = 0.0006). NGBf showed higher values when comparing the groups: NG (119 N), NGBf (129.8 N), T2D (105.8 N), and T2DBf (95.98 N) (Figure 3).

*Bending stress (MPa)*: The bending stress results were obtained to assess the femurs’ resistance to a given load applied during the bending test. There was a statistical difference in the NG vs. T2DBf (*p* = 0.0198), NGBf vs. T2D (*p* = 0.0130), and NGBf vs. T2DBf (*p* = 0.0006) group comparisons. Again, NGBf showed the highest values in the comparison: NG (36.43 MPa), NGBf (39.73 MPa), T2D (32.37 MPa), and T2DBf (29.48 MPa) (Figure 3).

*Bending displacement (mm)*: The bending displacement results evaluate bone deformation/displacement during the bending test. There were no statistical differences (*p* > 0.05) between the NG (0.8567 mm), NGBf (0.8583 mm), T2D (1 mm), and T2DBf (0.81 mm) groups (Figure 3).

*Modulus of elasticity (GPa)*: The elastic modulus is defined as the ratio between the applied stress and the deformity suffered by the sample during the bending test. The results showed no statistically significant differences (*p* > 0.05) between the NG (0.8233 GPa), NGBf (0.8917 GPa), T2D (0.7883 GPa), and T2DBf (0.7467 GPa) groups (Figure 3).

#### 2.3.2. Microtomography (MicroCT—Femurs)

*Bone volume percentage (BV/TV)*: The results of the micro-computed tomography of the bone volume percentage parameter showed no statistical difference (*p* > 0.05) between the NG (56.23%), NGBf (61.75%), T2D (59.1%), and T2DBf (48.94%) groups (Figure 4 and Figure 5).

*Trabecular thickness (TB.TH)*: The trabecular thickness results revealed no statistically relevant difference (*p* > 0.05) between the NG (0.1725 mm), NGBf (0.224 mm), T2D (0.18 mm), and T2DBf (0.1825 mm) groups (Figure 4 and Figure 5).

*Trabecular number (TB.N)*: When analyzing the number of trabeculae, there was a statistically significant difference only between the NG and NGBf groups (*p* = 0.0384). When comparing the NGBf groups, they showed the lowest number of trabeculae per mm: NG (3.355 1/mm), NGBf (2.64 1/mm), T2D (3.093 1/mm), and T2DBf (3.026 1/mm) (Figure 4 and Figure 5).

*Trabecular separation (TB.SP)*: The trabeculae separation results showed no statistical difference (*p* > 0.05) between the NG (0.164 mm), NGBf (0.18 mm), T2D (0.1667 mm), and T2DBf (0.174 mm) groups (Figure 4 and Figure 5).

*Percentage of cortical porosity PO (TOT)*: There was no statistically significant difference (*p* > 0.05) when comparing total porosity between the NG (45.58%), NGBf (39.88%), T2D (45%), and T2DBf (51.05%) groups (Figure 4 and Figure 5).

*Connectivity density (CONN.DN)*: The comparison between the connectivity density results showed a statistical difference only between the NG and NGBf groups (*p* = 0.0052). NG showed higher values compared to the following groups: NG (137.6 1/mm^3^), NGBf (73.96 1/mm^3^), T2D (90.56 1/mm^3^), and T2DBf (98.79 1/mm^3^) (Figure 4 and Figure 5).

### 2.4. Peri-Implantar Analyses

#### 2.4.1. Biomechanical Analyses (Removal Torque—N·cm)

The counter-torque results were higher in groups NG (6.98 N·cm) and NGBf (6.83 N·cm) compared to groups T2D (2.31 N·cm) and T2DBf (3.46 N·cm). There were statistically significant differences (*p* < 0.05) in the NG vs. T2D, NG vs. T2DBf, NGBf vs. T2D, and NGBf vs. T2DBf group comparisons. The results of the reverse torque indicate that systemic interference significantly affects bone metabolism in the event of an injury, suggesting that bone repair is more efficient in normoglycemic animals than in diabetic animals (Figure 6).

#### 2.4.2. Microtomography (MicroCT—Titanium Implants)

*Bone volume percentage (BV/TV)*: The results for percentage bone volume showed a statistically significant difference (*p* > 0.05) in the NG vs. T2D (*p* = 0.0003), NG vs. T2DBf (*p* = 0.0256), NGBf vs. T2D < 0.0001), NGBf vs. T2DBf (*p* = 0.0068), and T2D vs. T2DBf (*p* = 0.0394) group comparisons (Figure 7 and Figure 8).

*Trabecular thickness (TB.TH)*: For trabecular thickness, the groups with statistical differences were NG vs. T2D (*p* ≤ 0.0001), NGBf vs. T2D (*p* = 0.0001), and T2D vs. T2DBf (*p* = 0.0018). NG was the group with the highest score (0.1025), followed by NGBf (0.0965), T2DBf (0.0865), and T2D (0.05975) (Figure 7 and Figure 8).

*Trabecular number TB.N*: The results for the number of trabeculae showed a statistically significant difference (*p* > 0.05) when comparing the NG vs. T2D groups (*p* = 0.0042). The NG group had fewer trabeculae per mm: NG (4.497 1/mm), NGBf (3.54 1/mm), T2D (2.12 1/mm), and T2DBf (3.012 1/mm) (Figure 7 and Figure 8).

*Trabecular separation TB.SP*: There was a statistically significant difference when comparing the separation of trabeculae in the NG vs. T2D (*p* = 0.022) and NGBf vs. T2D (*p* = 0.0240) group comparisons. The group that obtained the highest result was T2D (0.126 mm), followed by T2DBf (0.119 mm), NGBf (0.1138 mm), and NG (0.1088 mm) (Figure 7 and Figure 8).

*Percentage of cortical porosity PO (TOT)*: The total porosity analysis showed a statistically significant difference between NG and T2D (*p* = 0.0049) and NGBf and T2D (*p* = 0.005). The T2D group had the highest percentage of total porosity (93.06%), followed by T2DBf (78.54%), NG (71.43%), and NGBf (64.23%) (Figure 7 and Figure 8).

*Bone surface (BS)*: When comparing the bone surface results, there was a statistically significant difference in the NG vs. T2D (*p* = 0.006), NGBf vs. T2D (*p* = 0.0051), and T2D vs. T2DBf (*p* = 0.0142) group comparisons. For this parameter, the group with the highest values was NG (21.17), followed by NGBf (17.25), T2DBf (16.42), and T2D (8.616) (Figure 7 and Figure 8).

*Intersection surface (IS)*: There was a statistical difference in the NG vs. T2D (*p* = 0.0032), NGBf vs. T2D (*p* = 0.0070), and T2D vs. T2DBf (*p* = 0.0076) group comparisons. When comparing the groups, we observed NG (9.327), NGBf (8.632), T2D (2.512), and T2DBf (9.045) (Figure 7 and Figure 8).

#### 2.4.3. Confocal Laser Microscopy Analysis

*Bone Dynamics/Fluorochrome Area*: A balance could be observed in the bone formation and mineralization processes in NG and NGBf, reflected in the similarity between the areas of calcein and alizarin red precipitation in these groups. In contrast, the T2D group showed little to no precipitation of mineral bone matrix throughout the osseointegration process, as evidenced by the smaller area of fluorochromes found in this group. The use of *Bauhinia forficata* in T2DBf partially restored the bone formation process in this animal type, characterized by an increase in the areas of fluorochrome precipitation when compared to the T2D group. The statistical differences are described in Table 3 (Figure 9 and Figure 10).

*Mineral Apposition Rate (MAR)*: Daily mineral apposition was calculated by measuring the fluorochrome precipitation lines by the difference between the days of injection (10 days). This provides the value, in mm, of the amount of mineralized bone matrix that was formed during the reported period. Thus, it can be observed that the administration of *Bauhinia forficata* in NGBf and T2DBf helped the bone mineralization process compared to the NG and T2D groups. After statistical analysis, there was a significant difference when comparing T2D vs. NG (*p* = 0.0014), NGBf (*p* = 0.0001), and T2DBf (*p* = 0.0033), where the T2D group had lower daily mineral apposition values (Figure 9 and Figure 11).

#### 2.4.4. Molecular Analysis (RT-qPCR)

*Vascular Endothelial Growth Factor (VEGF)*: The 14-day results revealed a statistically significant difference (*p* > 0.05) in the NG vs. T2DBf (*p* < 0.0001), NG vs. NGBf (*p* = 0.0026), NGBf vs. T2D (*p* = 0.0313), NGBf vs. T2DBf (*p* < 0.0001), and T2D vs. T2DBf (*p* < 0.0001) group comparisons. The group with the highest VEGF gene expression was T2DBf (5.273), followed by NGBf (1.852), T2D (1.188), and NG (1.009). At 28 days, there were statistical differences in the NG vs. T2DBf (*p* < 0.0001), NGBf vs. T2D (*p* = 0.0418), NGBf vs. T2DBf (*p* < 0.0001), and T2D vs. T2DBf (*p* < 0.0001) group comparisons. The groups with the highest scores were T2DBf (2.902), T2D (1.094), NG (1.005), and NGBf (0.8623) (Figure 12).

*Bone Sialoprotein (IBSP)*: At 14 days, there were no statistically significant differences between the groups. The group with the highest IBSP expression at 14 days was T2DBf (1.856), followed by NGBf (1.72), T2D (1.593), and NG (1.153). However, at 28 days, there were statistical differences in the NG vs. T2D (*p* = 0.0162), NG vs. T2DBf (*p* = 0.0027), NGBf vs. T2D (*p* = 0.0057), and NGBf vs. T2DBf (*p* = 0.0011) group comparisons. The group with the highest IBSP expression at 28 days was T2DBf (2.368), followed by T2D (1.937), NG (1.015), and NGBf (0.08703) (Figure 12).

*Osteocalcin (OCN)*: At 14 days, there were statistically significant differences in the NG vs. NGBf (*p* = 0.0041), NG vs. T2D (*p* < 0.0001), NG vs. T2DBf (*p* < 0.0001), NGBf vs. T2D (0.0384), and NGBf vs. T2DBf (*p* = 0.0468) group comparisons. The group with the highest OCN gene expression was NG (1.023), followed by NGBf (0.6164), T2DBf (0.3069), and T2D (0.2957). At 28 days, the groups with statistically significant differences for OCN were NG vs. T2D (0.0173), NG vs. T2DBF (0.0049), NGBf vs. T2D (0.0020), and NGBf vs. T2DBf (0.005). The group with the highest OCN expression was NGBf (1.237), followed by NG (1.067), T2D (0.3816), and, finally, T2DBf (0.2579) (Figure 12).

## 3. Discussion

One of the main risk factors for type 2 diabetes is an unhealthy lifestyle (poor diet, sedentary lifestyle, and obesity), and this is frequently found in today’s population, where the diet is rich in ultra-processed foods with low nutritional levels, therefore corroborating the significant increase in diabetes in recent years (PAHO/WHO) [31,32]. Projections for the future are alarming, with expectations of an exponential increase in the number of patients with T2D due to dietary and lifestyle patterns [33]. The combination of a poor diet and insufficient physical activity is leading to an increase in cases of diabetes, which represents a growing challenge for public health systems around the world. This scenario highlights the urgent need for effective interventions to combat the type 2 diabetes epidemic, including strategies to promote lifestyle changes and the development of complementary therapeutic approaches, such as the use of herbal medicines to improve metabolic health and bone quality [17].

The results indicate that *Bauhinia forficata* is very promising in terms of improving the bone microstructure and mechanical properties of long bones that have undergone injury [19]. Although type 2 diabetes has negative effects on bone metabolism, this study found little direct evidence of these effects in uninjured long bones [34]. It was observed that type 2 diabetes is more harmful in the presence of implants, with worse peri-implant osseointegration. The systemic treatment with Bf showed effectiveness in weight homeostasis and in reducing glycemic levels. It also improved bone biomechanics, quantity, and quality in implant sites. In the first three weeks before the STZ injection, the groups given the T2D and T2DBf cafeteria diet already showed a significant increase in weight, indicating that the hypercaloric diet, combined with a sedentary lifestyle, is a factor that can lead to insulin resistance. When STZ was injected, noticeable weight loss was observed in the untreated diabetic animals, although it is also worth noting that the treated diabetic animals weighed significantly less. This suggests that *Bauhinia forficata* prevented the weight loss typically associated with STZ-induced β-cell toxicity, helping maintain weight stability in the diabetic animals. In contrast, the gradual increase in weight observed in the normoglycemic animals can be attributed to the stress experienced during this study. This stress raised circulating cortisol levels, which then led to an increase in visceral adiposity [4].

The mechanism of action of *Bauhinia forficata* has not yet been fully explained, but it is attributed, in particular, to the presence of flavonoids and lactones, which act by modulating the activity of pancreatic beta cells and improving insulin sensitivity [24,26]. In addition, *Bauhinia forficata* appears to have an antioxidant effect, contributing to a lower level of oxidative stress, a critical factor in the pathogenesis of type 2 diabetes [25]. These combined effects show that Bf could be a valuable alternative for glycemic control and management of type 2 diabetes, offering a natural approach to conventional therapies. When comparing the diabetic groups, it was possible to observe that from the moment the tea was administered (day 28), the fasting blood glucose levels of these animals began to decrease until euthanasia, showing that the herbal medicine improves insulin sensitivity in type 2 diabetic rats. The use of herbal treatment was not able to re-establish normoglycemia, but it did obtain optimistic results for the use of this natural drug. One hypothesis is that the study period was not long enough for *Bauhinia forficata* to be fully effective. On the other hand, the NG and NGBf animals maintained similar glycemic results, indicating that *Bauhinia forficata* has no hypoglycemic effect on normoglycemic animals.

The flavonoid Kaempferol, especially its 3,7-di-O-α-L-rhamnoside form, plays an important role in the hypoglycemic effects of Bf because it has the ability to eliminate free radicals, contributing to the protection of cells against oxidative stress and inflammation [14,15,16,17,18,19,20,21,22,23,24,25]. Besides this effect, it modulates the cell cycle by influencing nuclear factor Kappa-B (NF-kB) [14], which affects the OPG/RANK-L pathway, inactivating osteoclasts and increasing osteoblast proliferation through the expression of alkaline phosphatase [35]. This dual effect on both glycemic control and bone tissue modulation justifies the use of this natural compound. Correlating the values of the three-point bending test, the NGBf group had the highest maximum breaking force compared to the other groups, even the NG group, showing that the bone needed more force in Newtons to break and, therefore, had greater bone rigidity [36]. Furthermore, there was a statistically significant difference between the NGBf and T2D Bf groups. When observing the modulus of elasticity, it was clear that the NGBf group also obtained the best results; it showed the highest modulus of elasticity, suggesting that it had the most resilient bone. Regarding the bending displacement parameter, the group with the least deformation was T2DBf, followed by NGBf, thus characterizing the groups with the best quality in this parameter. Finally, in the bending stress analysis, the NGBf group had the highest value; therefore, the force in MPa was the greatest for bending to occur. From all the EMIC data, it was possible to see that the NGBf group had better bone strength and stiffness. Computed microtomography of the femoral neck corroborated the EMIC results, showing that in long bones that had not been injured, neither the metabolic disorder nor the treatment with herbal medicine affected bone microarchitecture. However, the microarchitecture of the tibiae of the normoglycemic animals was much better than that of the diabetic animals, and, in addition, when comparing the T2D and T2DBf groups, an improvement in the quantity and quality of bone was observed in the diabetic animals treated with *Bauhinia forficata* [37,38].

Although the three-point bending test revealed no statistically significant differences among the groups, this finding reinforces that *Bauhinia forficata* administration did not compromise the mechanical integrity of uninjured long bones. Likewise, under noninjury conditions, type 2 diabetes did not significantly alter the biomechanical performance or metabolic characteristics of the femur. In contrast, when evaluating the tibiae, microtomographic analyses demonstrated a clear deterioration in trabecular architecture and bone quality, particularly in the diabetic animals without treatment. This result confirms that diabetes effects on bone metabolism are more pronounced in reparative or stress-induced conditions.

To assess *Bauhinia forficata*’s ability to accelerate osteoinductive properties, the expression of osteogenic genes was quantified by qRT-PCR after 14 and 28 days. The marker genes analyzed included vascular endothelial growth factor (VEGF), bone sialoprotein (IBSP), and osteocalcin (OCN). All these genes were upregulated in the samples from the animals treated with *Bauhinia forficata*, showing that systemic treatment had a beneficial effect on osteogenic differentiation.

Based on the RT-PCR results, vascular endothelial growth factor (VEGF) is an essential element for maintaining bone homeostasis, since it can stimulate the differentiation of mesenchymal stem cells into osteoblasts based on an intrathecal mechanism and can also stimulate the differentiation of monocytes into osteoclasts based on a paracrine mechanism [39,40]. *Bauhinia forficata* treatment in the T2DBf group led to an increase in VEGF expression after both 14 and 28 days, with the group showing the highest expression of this gene in the two periods analyzed, which can be interpreted as greater angiogenic induction. It is important to note that VEGF exhibits a biphasic function during bone repair. Its transient increase in early stages favors angiogenesis and osteoblast recruitment, while persistent expression at advanced stages may reflect delayed remodeling. In the present study, the elevated VEGF levels observed in the T2DBf group at both 14 and 28 days should not be interpreted as pathological but rather as sustained pro-angiogenic activity supporting bone maturation. This interpretation aligns with the concomitant improvements in microtomographic and confocal analyses, which demonstrated enhanced mineral apposition and trabecular organization in the same group. IBSP is a structural protein that specifically expresses fully differentiated osteoblasts, indicating an already mature and fully mineralized tissue. The group that expressed this protein most strongly at 14 and 28 days was T2DBf, indicating that systemic treatment with Bf plays a crucial role in osteogenic differentiation. Osteocalcin (OCN) is a mineralization gene found in the later stages of osteogenic differentiation [41]. After 14 days, the group with the highest expression was NG, followed by the NGBf group. After 28 days, this was reversed, with the NGBf group playing the leading role in the expression of this gene, indicating rapid cell induction and differentiation. The downregulation of osteocalcin at 28 days in the T2DBf group may indicate a temporal shift in osteogenic maturation rather than reduced bone formation. Considering that OCN is a late marker of matrix mineralization, its lower expression likely reflects a transition from active transcriptional activity to extracellular protein accumulation. The confocal microscopy data support this interpretation, demonstrating continued mineral apposition and deposition of fluorochromes in the same group, confirming the progression of bone formation despite reduced gene expression. Therefore, the RT-PCR results confirm that osteogenic differentiation was increased by the systemic use of the herbal medicine *Bauhinia forficata*.

The laser confocal microscopy results were obtained from the precipitation of fluorochromes (calcein and alizarin), which are related to mineralization in bone tissue [10,42]. This analysis showed a balance in the process of bone formation and mineralization among the NG and NGBf groups, as evidenced by the similarity in the areas of calcein and alizarin precipitation in these groups. However, the T2D group showed limited bone mineral matrix precipitation during the osseointegration process, as evidenced by the low amount of fluorochromes precipitated in both periods. But when observing fluorochrome precipitation in the group that received systemic administration of the herbal medicine (T2DBf), a partial restoration of the bone formation process was noted, as evidenced by the increase in fluorochrome precipitation compared to the untreated diabetic group. This is related to the low expression of osteocalcin seen in the PCR analysis, since the protein was expressed in the bone tissue of diabetic animals. Nonetheless, this fact can be better proven by immunohistochemical analysis, which aims to analyze the processes of osteoblastic differentiation, mineralization, and bone resorption.

Regarding the daily mineral deposition rate, it was observed that the groups in which *Bauhinia forficata* was administered (NGBf and T2DBf) obtained a greater amount of mineralized bone matrix deposition in mm in the period between the calcein and alizarin red injections. These confocal microscopy results corroborate the expression of the IBSP gene, showing that after 14 days of euthanasia, the NGBf and T2DBf groups were the ones that expressed the most bone sialoprotein, leading us to believe that the bone tissue in this group reached mineralization maturity and should begin to undergo bone remodeling [43].

Dental procedures such as tooth extraction and implant installation cause damage to bone tissue, which can be particularly worrying when diabetic patients need to undergo these types of procedures [10,11,44]. Osseointegration, a critical process for the success of dental implants, is greatly compromised in diabetic individuals a priori by factors such as impairment of the healing process, leading to implant failure [45]. A decrease in bone quality was demonstrated by the removal torque values of the T2D group, and the concentration of *Bauhinia forficata* improved the mechanical characteristics of the bone, although it was not effective in reversing this response as in the NGBf animals. Regarding computerized microtomography of the tibia, the results also confirmed what was seen in the counter-torque, suggesting that diabetes damages osseointegration and *Bauhinia forficata* improves it. The NG group obtained superior results to the T2D group, and, furthermore, systemic interference is highly relevant to bone metabolism when a tissue injury occurs. These observations indicate that in the NG group, the healing phases progress smoothly, including hemostasis, inflammation, proliferation, and remodeling. In the T2D group, the inflammatory phase is predominant and prolonged, resulting in a dysfunctional repair process. Similar results to the NG group were observed between the groups treated with the tea (NGBf and T2DBf). Finally, when comparing the T2D group with the T2DBf group, it was observed that the microarchitecture of the treated animals was better than that of the untreated animals, in terms of bone volume, bone surface, and intersection and trabecular thickness, resulting in better repair.

From a translational perspective, the findings suggest that *Bauhinia forficata* may serve as an adjuvant strategy for managing bone repair complications in diabetic individuals. The dose administered (50 g/L infusion) corresponds to the concentration recommended by FIOCRUZ [46], which indicates the use of 35 mg of dried leaves for 120 mL of water in traditional preparations, 2–3 times per day as an adjuvant of glycemic control [47]. In our experimental design, this proportion was converted to a liter-based measurement to facilitate preparation and ensure homogeneous daily availability for the animals. Therefore, *B. forficata* should be considered a complementary, rather than primary, therapeutic approach, potentially aiding perioperative metabolic stability and osseointegration in diabetic patients. Future studies should investigate pharmacokinetics, long-term safety, and possible synergistic effects with standard antidiabetic therapies to support its clinical applicability.

A limitation of this study is the absence of direct phytochemical quantification of the specific *Bauhinia forficata* batch used. Although the composition of *Bf* infusions has been extensively characterized in the literature [46,47,48,49,50], future studies should include direct chemical profiling of the administered batch to confirm the concentration of bioactive compounds such as kaempferitrin and quercetin.

## 4. Materials and Methods

This study was approved by the Animal Use Ethics Committee (CEUA) of São Paulo State University (UNESP), Araçatuba School of Dentistry, Brazil (approval number 0129-2021, approved on 24 June 2021). This study followed the guidelines for Reporting In Vivo Experiments (ARRIVE) and the standards of the Guide for the Care and Use of Laboratory Animals of the US National Institutes of Health (Institute of Laboratory Animal Resources) [51].

### 4.1. Animals

Thirty-two male *Wistar* rats (*Rattus norvegicus albinus*), weighing 250 g and 3 months old, were divided into four groups (n = 8 in each group): normoglycemic (NG), normoglycemic + *Bauhinia forficata* (NGBf), type 2 diabetic (T2D), and type 2 diabetic + *Bauhinia forficata* (T2DBf). The animals were identified by number and randomly divided using Microsoft Office Excel software (Microsoft, Redmond, WA, USA), maintaining a 1:1 allocation ratio for each group.

The animals were kept in cages in a temperature-controlled environment (22 ± 2 °C; controlled light cycle with 12 h of light and 12 h of darkness). The normoglycemic animals (NG and NGBf) were fed a conventional diet throughout the experiment (NUVILAB, 1.4% Ca and 0.8% P + water ad libitum). The animals intended to be diabetic were started on a cafeteria diet, which will be described in the next section.

### 4.2. Type 2 Diabetes Induction

In order to induce type 2 diabetes, a cafeteria diet associated with the administration of low-dose streptozotocin (STZ) was used only in the T2D and T2DBf groups. The cafeteria diet is a high-fat diet that simulates a nutrient-poor meal, promoting insulin resistance, with high levels of lipids, carbohydrates, and sugars [10,52,53].

During the first three weeks, each animal received 30 g of this diet, consisting of 10 g of corn snacks, 10 g of wafer crackers, and 10 g of stuffed crackers, as well as sugar water with 12% sucrose (50 mL/day). On day 21, STZ application was performed as follows: the animals in the T2D and T2DBf groups were anesthetized with xylazine (5 mg/kg) and ketamine (50 mg/kg), disinfected with 70% alcohol in the scrotal region, and injected with STZ (35 mg/kg) into the penile vein, dissolved in 0.1 M citrate buffer (pH 4.5), following the same anesthetic and dosage protocol stipulated in Pitol-Palin et al. 2021 [10]. The normoglycemic animals received only the vehicle to undergo the same surgical stress. The animals in the normoglycemic groups received only the vehicle to undergo the same surgical stress.

STZ has a toxic effect on pancreatic β-cells and, at low doses, induces partial dysfunction in animal bodies. This means that these animals still produce insulin, but not optimally. We considered the following periods: the start of the diet, application of STZ, confirmation of type 2 diabetes and beginning of the treatment with tea, implant surgery, and euthanasia. The combination of a cafeteria diet and STZ resulted in the development of type 2 diabetes, which was confirmed 7 days after induction, when the animals had to have blood glucose levels above 198 mg/dL to be considered diabetic [10,34]. To carry out the glycemic analysis, the animals were fasted for 2 h and, after this phase, a blood sample was taken from the tail ban through a small incision with a No. 11 scalpel blade, which was sufficient for reading on the glucometer. This procedure was performed weekly from day 0 onwards [30].

### 4.3. Bauhinia forficata Link Tea

After T2D confirmation (7 days after STZ application), the systemic treatment with *Bauhinia forficata* tea was started for the animals in the NGBf and T2DBf groups. This treatment used 50 g of dried *Bauhinia forficata* (Aphoticario Farmacia de manipulação Ltd.a, Araçatuba, SP, Brazil, Batch: PAV 0424/2713-ROS) leaf infused in 1 L of water. The dose was established taking into account the ratio established by FIOCRUZ, which recommends 5 g of leaves in 100 mL of water [46]. The cold tea was provided daily in bottles designed specifically for these animals, replacing pure potable water ad libitum in the cages until the end of the experiment [17]. Each cage received a bottle of tea, and its volume was measured and replenished daily (Table 4).

In our study, we deliberately opted to use the dried leaves in the form of an infusion, in order to mimic the most common consumption method in the population, which traditionally prepares the tea from dried leaves. This approach reflects the real-world use of the plant rather than concentrated organic extracts. Additionally, it is important to highlight that very few studies describe the phytochemical composition of *Bauhinia forficata* leaves in detail.

#### *Bauhinia forficata* Link Tea Chemical Profile

The chemical profile of *Bauhinia forficata* Link was verified based on previous studies in the literature [46,47,48,49,50]. As the main objective and focus of this study is its translational aspect, transforming the treatment performed by the population into an experimental model in animals, we conducted a bibliographic overview of the main compounds and active elements found both in the leaves of *Bauhinia forficata* Link and in the infusion made from its leaves.

*Bauhinia forficata* is dominated by flavonoids, phenolic acids, and terpenoids. High-resolution chromatographic techniques, such as HPLC-DAD-ESI/MSⁿ and HPLC-PDA-ESI-IT-MSn, have identified up to 39 distinct flavonoids, with kaempferitrin (kaempferol-3,7-di-O-α-L-rhamnopyranoside) being recognized as the chemical marker of the species [48,49]. Several flavonoids have been identified in the leaves of *Bauhinia forficata*, mainly derivatives of quercetin and kaempferol. Among them, isorhamnetin-3-O-glucoside (C_22_H_22_O_12_), quercetin-O-hexoside (C_21_H_19_O_12_), quercetin-3-O-α-L-rhamnopyranoside (C_21_H_20_O_11_), isoquercetin (C_22_H_22_O_12_), quercetin-3-O-(2-rhamnosyl)rutinoside (C_34_H_42_O_19_), and quercetin-3-O-rutinoside (C_27_H_30_O_16_) have been described. In parallel, kaempferol derivatives include kaempferol-3-rhamnoside (C_21_H_20_O_10_), kaempferol-3-O-α-L-rhamnoside (C_21_H_20_O_10_), kaempferol-7-O-glucoside (C_22_H_22_O_11_), kaempferol-3-O-(4-O-p-coumaroyl)glucoside (C_30_H_26_O_13_), kaempferol-3-O-rutinoside (C_27_H_30_O_15_), kaempferol-3-O-(2-rhamnosyl)rutinoside (C_33_H_40_O_20_), and, particularly, kaempferitrin (C_27_H_30_O_14_), which is recognized as the chemical marker of the species. Other reported compounds include 3,7-di-O-α-rhamnopyranosyl quercetin (C_27_H_30_O_15_), rutin (C_27_H_30_O_16_), and quercetin-3-rutinoside-7-rhamnoside. Beyond flavonoids, phenolic compounds, such as trans-caffeic acid (C_9_H_8_O_4_), and miscellaneous metabolites, such as HY52 (C_17_H_30_N_2_O_2_), have also been identified. Considering this abundance of phenolic metabolites, *Bauhinia forficata* extracts are expected to exhibit strong antioxidant activity. Not surprisingly, the species is traditionally referred to as “natural insulin” and has been extensively investigated for its hypoglycemic and antioxidant potential [50].

Phytochemical characterization of *Bauhinia forficata* infusions has revealed a high content of phenolic metabolites, mainly flavonoids, with a total phenolic content (TPC) ranging from 1923 to 6355 mg RE/100 g and a total flavonoid content (TFC) from 482 to 3700 mg RE/100 g [46,47,48,49,50]. Significant variability among commercial batches was observed, highlighting the influence of cultivation, harvesting, and processing practices on bioactive compound stability. LC-HRMS analysis tentatively identified 20 phenolic compounds, predominantly kaempferol and quercetin derivatives, such as rutin, isoquercetin, quercetin-3-O-rhamnoside, and kaempferol-3-O-glucoside, corroborating previous reports on the pharmacological relevance of these flavonoids. Volatile fraction analysis by HS-SPME/GC–MS revealed 40 compounds, mostly sesquiterpenes (e.g., spathulenol and caryophyllene oxide), along with alcohols, esters, and norisoprenoids, many of which display antioxidant, anti-inflammatory, and antimicrobial activities. Notably, all infusions inhibited α-amylase activity, with IC_50_ values ranging from 0.235 to 0.801 mg RE/mL, supporting their traditional use in the management of postprandial hyperglycemia, despite being less potent than acarbose (IC_50_ = 0.034 mg/mL). Since these compounds have been repeatedly reported and quantified in previously validated analytical protocols, we used this bibliographic evidence to ensure translational relevance and reproducibility of the preparation, maintaining our focus on the biological and preclinical outcomes rather than on re-characterization of the extract composition. Together, these findings reinforce that *Bauhinia forficata* infusions constitute a polyphenol-rich phytocomplex with relevant antioxidant and hypoglycemic properties, consistent with its popular designation as “natural insulin” [46,47,48,49,50].

### 4.4. Implant Installation

Implant placement surgery was performed 14 days after the beginning of the tea treatment. The animals were fasted for 8 h and then sedated for antisepsis and trichotomy, as described previously [54].

Each animal received one implant in each tibia, totaling 64 implants, which were 2.0 mm wide and 4.0 mm long (Titaniumfix, São José dos Campos, SP, Brazil) and sterilized by gamma rays. A 1.5 cm incision was made in the tibial metaphysis to allow surgical access. After soft tissue dissection, the bone surface was exposed for implant placement. Implants were inserted bilaterally in the tibiae with bicortical fixation, according to the assigned groups. Closure was performed using Vicryl sutures, and immediately after surgery, all animals received pentabiotic and sodium dipyrone [54,55,56,57].

### 4.5. Euthanasia and Proposed Analyses (Sample Processing)

The animals were euthanized by anesthetic overdose to collect femurs and tibiae (peri-implant evaluation) at 14 and 28 days post-operative. At 14 days, RT-qPCR analysis was performed on the collected peri-implant bone (tibia) samples. At 28 days, bone biomechanics assessment tests (3-point test on the femurs and removal torque on the tibias with implants), MicroCT (femurs and peri-implant region), confocal laser microscopy analysis (peri-implant region), and RT-qPCR (peri-implant region) were performed. Right tibias were used to evaluate removal torque, and after implant removal, the peri-implant bone was collected for PCR. Microtomographic and confocal microscopy analyses were performed on the left tibiae. In addition, the right femurs were collected for 3-point tests on the femurs, while the left femurs were sent for microtomographic analysis.

#### 4.5.1. Three-Point Bending Testing (EMIC)

The EMIC tests performed on the femurs aimed to evaluate the biomechanical characteristics of long bones without interference from surgical procedures, such as the installation of implants. The EMIC/Mod DL3000 Universal Testing Machine was used, in which the femurs were positioned horizontally on a platform with two supports on the machine, at a distance of 20 mm, and a load cell capacity of 2000 N, as described in Wajima et al. 2024 [34]. We evaluated the following parameters: maximum breaking force (N), bending stress (MPa), bending displacement (mm), and modulus of elasticity (GPa). The values were recorded by a software program installed on a computer connected to the equipment, directly providing all the values presented in this study [34,36].

#### 4.5.2. Microtomography (MicroCT)

The tibia and femur samples were fixed in formaldehyde for 48 h, washed in running water for 24 h, immersed in 70% alcohol, and then scanned using computerized microtomography equipment.

Femoral *Micro-CT* [34,58]: The samples were scanned by SkyScan 1272 microtomograph (SkyScan 1272 Bruker-MicroCT, Kontich, Belgium), reconstituted, corrected, and evaluated according to Wajima et al. 2024 [34]. Then, using CTAnalyser—CTAn software (2003-11SkyScan, 2012 Bruker MicroCT Version 1.12.4.0), two regions of interest (ROI) were standardized in the distal portion of the femur for trabecular bone: (I) upper boundary, 100 slices below the growth plate, and (II) lower boundary, 70 slices from the upper boundary (90 µm). In the trabecular portion, the bone volume percentage (Tb. BV/TV), structure model index (Tb. SMI), trabecular number (Tb. N), trabecular thickness (Tb. Th), and trabecular separation (Tb. Sp) were analyzed.

*Micro-CT of the peri-implant area* [54,58,59]: The samples were scanned by SkyScan 1272 microtomograph (SkyScan 1272 Bruker-MicroCT, Belgium), reconstituted, corrected, and evaluated according to Pitol-Palin et al. 2025 [59]. Thus, we were able to define the percent bone volume (BV/TV), trabecular thickness (Tb. Th), trabecular number and separation (Tb. N, Tb. Sp), and bone-to-implant contact (BIC) through the intersection surface. In the trabecular portion, the percentage of bone volume (BV/TV), the number of trabeculae (Tb. N), the thickness of the trabeculae (Tb. Th), the separation of the trabeculae (Tb. Sp), and the connectivity density (Conn. Dn) were analyzed.

#### 4.5.3. Peri-Implant Biomechanic (Removal Torque)

The animals were sedated using the same anesthetic protocol described above [30]. The animals’ right tibiae were accessed so that the implant region could be exposed. The right tibias were exposed for visualization of the implants. Using a digital torque wrench (Homis, São Paulo, SP, Brazil) attached to a hexagonal wrench, counterclockwise rotation was performed until complete rupture of the bone/implant interface occurred. The maximum rupture value (N·cm) was recorded [54,59].

#### 4.5.4. Confocal Laser Microscopy

Following MicroCT analysis, the specimens were dehydrated in graded ethanol and infiltrated with a 1:1 mixture of methyl methacrylate and acetone. They were then subjected to three PMMAL baths, the last containing 1% benzoyl peroxide as a catalyst, and polymerized at 37 °C for 5 days in sealed containers. After polymerization, the blocks were trimmed with a bench motor and further polished manually with an automatic system (ECOMET 250PRO/AUTOMET 250, Buehler, Lake Bluff, IL, USA). Sections with 20 mm were mounted on histological slides and examined under a Leica CTR 4000 CS SPE microscope at 10× magnification. The reconstructed images revealed overlapping fluorochromes in the alveolar bone, and ImageJ software 1.54d (Image Processing and Analysis Software) was used to quantify bone dynamics based on the distinction between old and newly formed bone [10,42].

#### 4.5.5. Molecular Analysis (RT-qPCR)

For PCR, peri-implant bone fragments were collected at two times of euthanasia. The specimens were carefully washed in PBS and then frozen in liquid nitrogen so that the total RNA could be extracted using the Trizol reagent (Life Technologies: Invitrogen, Carlsbad, CA, USA) and kept in a freezer at −80 °C until analysis. After checking the purity and concentration of the RNA, 1 µg of RNA was synthesized using the reverse transcriptase reaction with M-MLV reverse transcriptase (Promega Corporation, Madison, WI, USA). The cDNAs obtained were pipetted, together with the Taqman Fast Advanced Mastermix (Applied Biosystems, Waltham, MA, USA), into a PCR plate (96-well fast thermal cycling, Life Biotechnologies), for the detection of genes involved in collagen metabolism and the bone repair process, using Taqman Gene Expression Assays. The targeted genes for study included VEGF, associated with vascularization, plus OCN and BSP, related to the extracellular matrix. Real-time PCR was conducted on the Step One Plus detection system (Applied Biosystems) under the following conditions: 50 °C for 2 min, 95 °C for 10 min, followed by 40 cycles of 95 °C for 15 s and 60 °C for 1 min, with a standard denaturation curve at the end. The relative gene expression was calculated based on the expression of mitochondrial ribosomal proteins and normalized according to the gene expression of the bone fragments from the alveoli in different experimental periods, using the ΔΔCT method. This study was carried out in quadruplicate [60,61]. 

#### 4.5.6. Statistical Analysis

GraphPad Prism 7.03 software (GraphPad Software, La Jolla, CA, USA) was used for statistical analysis. To confirm a normal distribution, the Shapiro–Wilk test was used in order to verify the homoscedasticity in the data. The one-way ANOVA test was used to determine whether there were significant differences between the groups. For direct comparisons between the groups, the Holm–Sidak post hoc test was applied. A significance level of 5% was applied to all analyses.

## 5. Conclusions

Systemic use of *Bauhinia forficata* has a beneficial effect on the etiopathogenesis of type 2 diabetes, promoting weight stabilization and helping to control animals’ glycemic levels, bringing them closer to normoglycemia. Although the systemic condition of type 2 diabetes did not alter the structural characteristics of uninjured long bones (femurs), the presence of T2D induced negative effects on osseointegration in response to tissue injury (installation of implants in the tibia), resulting in a reduction in implant removal strength and damage to bone microarchitecture. Daily treatment with *Bauhinia forficata* in animals with type 2 diabetes (T2DBf) improved peri-implant biomechanics, although it was not enough to restore the reparatory characteristics observed in the control groups, such as NG and NGBf, and also improved the characteristics of bone quantity and quality around the implants. Importantly, the use of *Bauhinia forficata* in normoglycemic rats caused no adverse effects.

## Figures and Tables

**Figure 1 pharmaceuticals-18-01724-f001:**
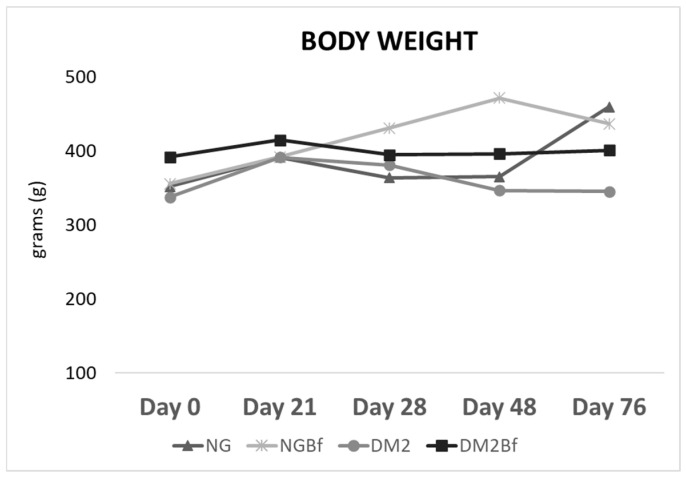
Graph showing the body weight of the NG, NGBF, T2D, and T2DBf groups. The *x*-axis represents the days on which body weight was measured, and the *y*-axis indicates the mean body weight (grams).

**Figure 2 pharmaceuticals-18-01724-f002:**
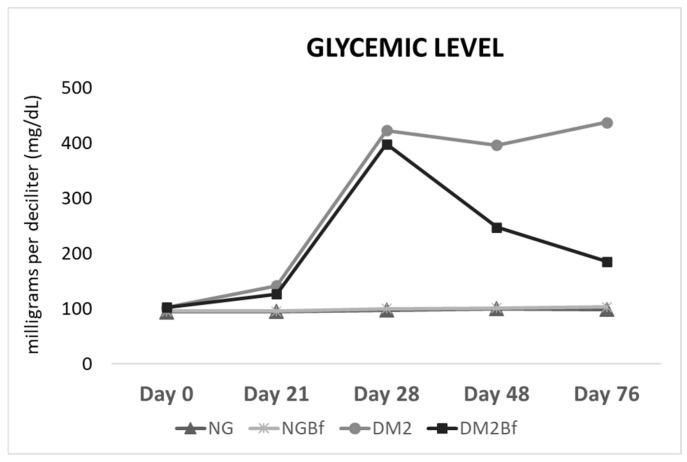
Graph showing the glycemic level for the NG, NGBF, T2D, and T2DBf groups. The *x*-axis represents the days on which glycemic levels were measured, and the *y*-axis indicates the mean blood glucose values (mg/dL).

**Figure 3 pharmaceuticals-18-01724-f003:**
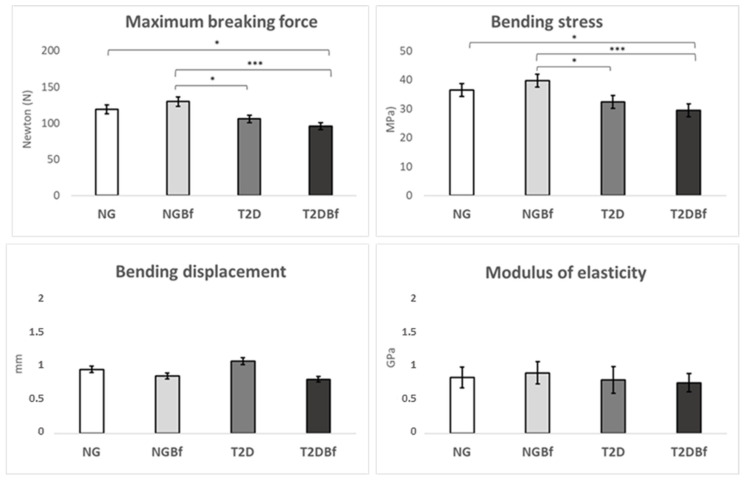
Graphs showing the three-point testing. The *x*-axis shows the groups evaluated in the study. The *y*-axis shows the values for force (N), megapascals (MPa), millimeters (mm), or gigapascals (GPa). The results of the one-way ANOVA statistical test (*p* < 0.05) are marked with brackets, where the higher the number of asterisks (* or ***), the higher the difference between the groups.

**Figure 4 pharmaceuticals-18-01724-f004:**
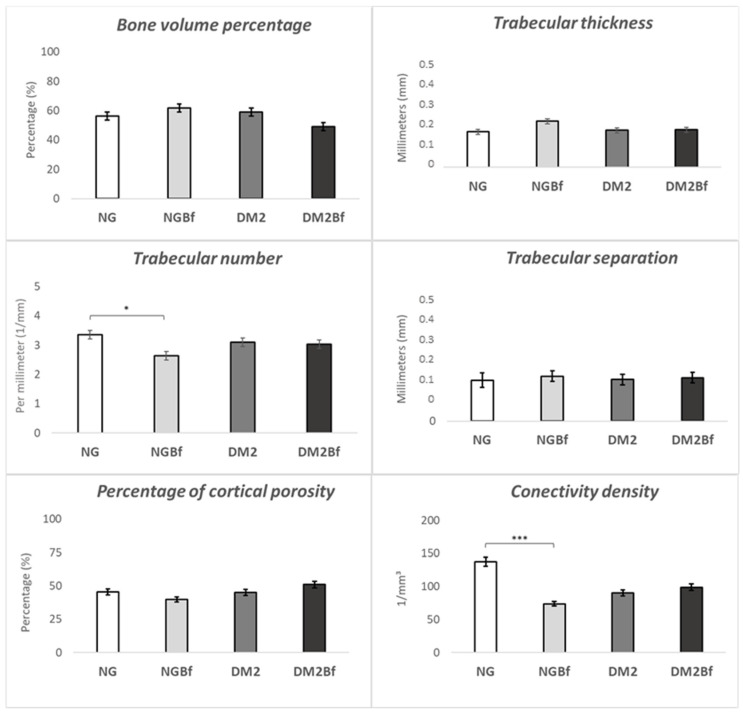
Microtomographic analysis of the comparison between NG, NGBf, T2D, and T2DBf on the femur. The *x*-axis shows the groups evaluated in the study. The *y*-axis shows the values obtained on microtomographic analysis. The results of the one-way ANOVA statistical test (*p* < 0.05) are marked with brackets, where the higher the number of asterisks (* or ***), the higher the difference between the groups.

**Figure 5 pharmaceuticals-18-01724-f005:**
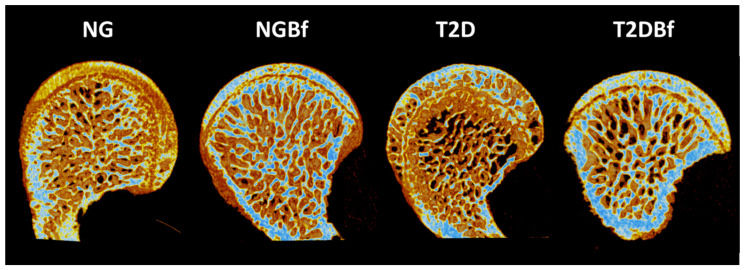
Three-dimensional reconstruction of the femoral head region of the NG, NGBf, T2D, and T2DBf groups. The evaluation was performed on the femoral head region, where it was observed that the prevalence of type 2 diabetes or treatment with *Bauhinia forficata* did not interfere with bone metabolism without injury.

**Figure 6 pharmaceuticals-18-01724-f006:**
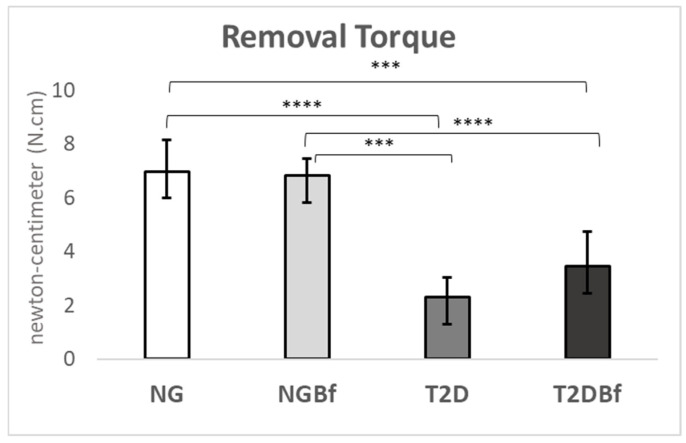
Graphs showing biomechanical analysis in the peri-implant region. The normoglycemic groups (NG and NGBf) showed an impaired biomechanical value when compared to the type 2 diabetes groups (T2D and T2DBf). The results of the one-way ANOVA statistical test (*p* < 0.05) are marked with brackets, where the higher the number of asterisks (*** or ****), the higher the difference between the groups.

**Figure 7 pharmaceuticals-18-01724-f007:**
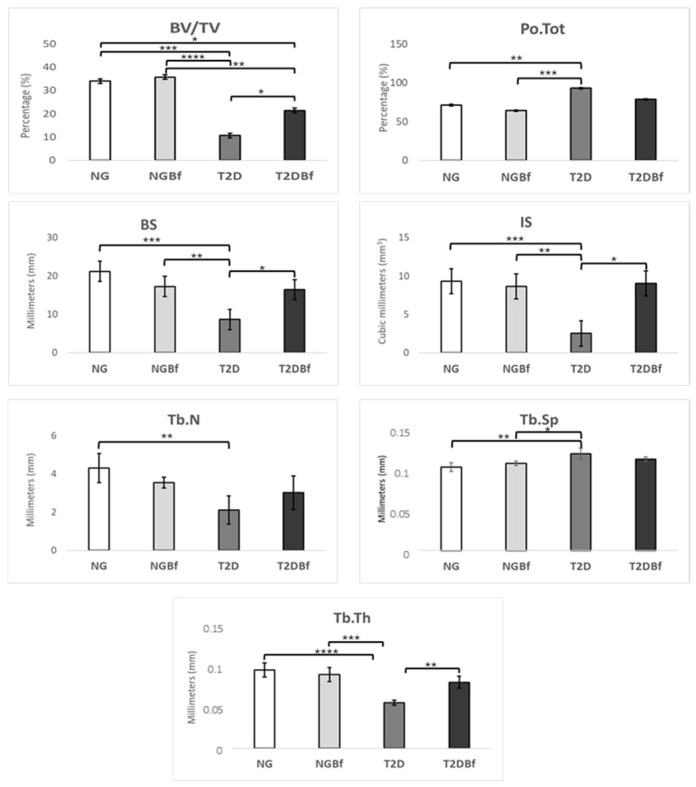
Microtomographic analysis of the comparison between NG, NGBf, T2D, and T2DBf in the peri-implant region. The *x*-axis shows the groups evaluated in this study. The *y*-axis shows the values obtained on microtomographic analysis. The results of the one-way ANOVA statistical test (*p* < 0.05) are marked with brackets, where the higher the number of asterisks (*, **, ***, or ****), the higher the difference between the groups.

**Figure 8 pharmaceuticals-18-01724-f008:**
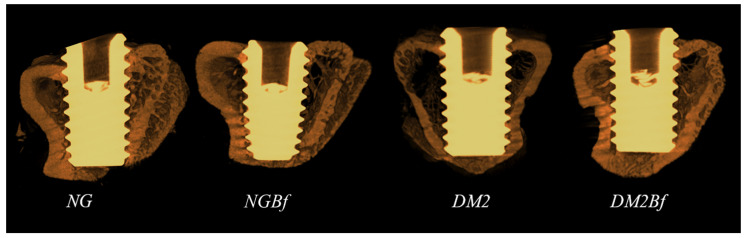
Images obtained by CTVox software (Bruker MicroCT Version 1.12.4.0) through microtomographic analysis of the peri-implant region in the NG, NGBf, T2D, and T2DBf groups. The evaluation was performed at the implant region. In contrast to the microtomography findings for the femurs, it was noted that type 2 diabetes impaired reparative bone metabolism, while the use of *Bauhinia forficata* promoted improvement in the parameters of peri-implant bone microarchitecture.

**Figure 9 pharmaceuticals-18-01724-f009:**
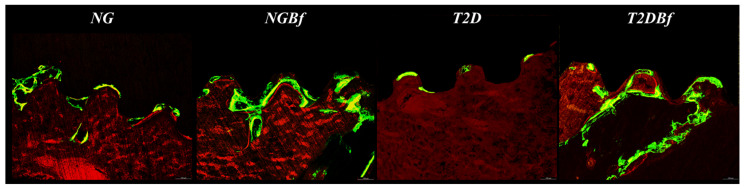
Photomicrographs representing the NG, NGBf, T2D, and T2DBf groups by confocal laser microscopy. It is possible to note the low mineral matrix deposition in the type 2 diabetes group, while treatment with *Bauhinia forficata* supported bone dynamics and mineral apposition. In green: Calcein precipitation in the bone matrix. In red: Alizarin red precipitation in the bone matrix. Original 10× magnification.

**Figure 10 pharmaceuticals-18-01724-f010:**
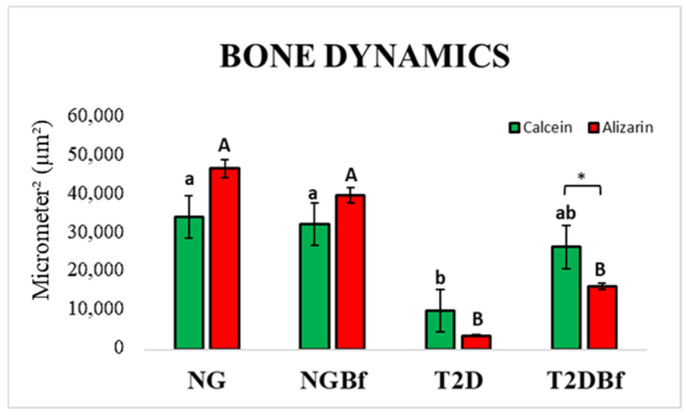
Bone dynamics by laser confocal microscopy in the NG, NGBf, DM2, and DM2Bf groups. Different letters indicate statistically significant differences between fluorochromes, * indicate statistically significant differences in the group. One-way ANOVA statistical test (*p* < 0.05).

**Figure 11 pharmaceuticals-18-01724-f011:**
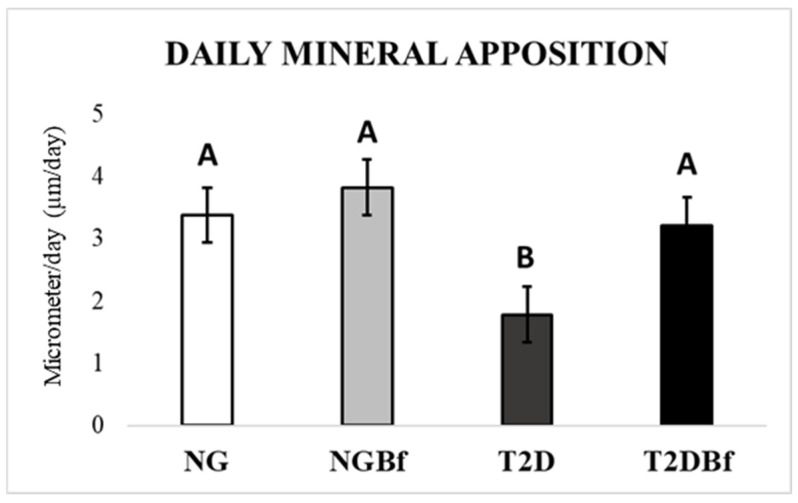
Mineral apposition rate by confocal laser microscopy in the NG, NGBf, T2D, and T2DBf groups at 28 days of peri-implant repair. Different letters indicate statistically significant differences. One-way ANOVA statistical test (*p* < 0.05).

**Figure 12 pharmaceuticals-18-01724-f012:**
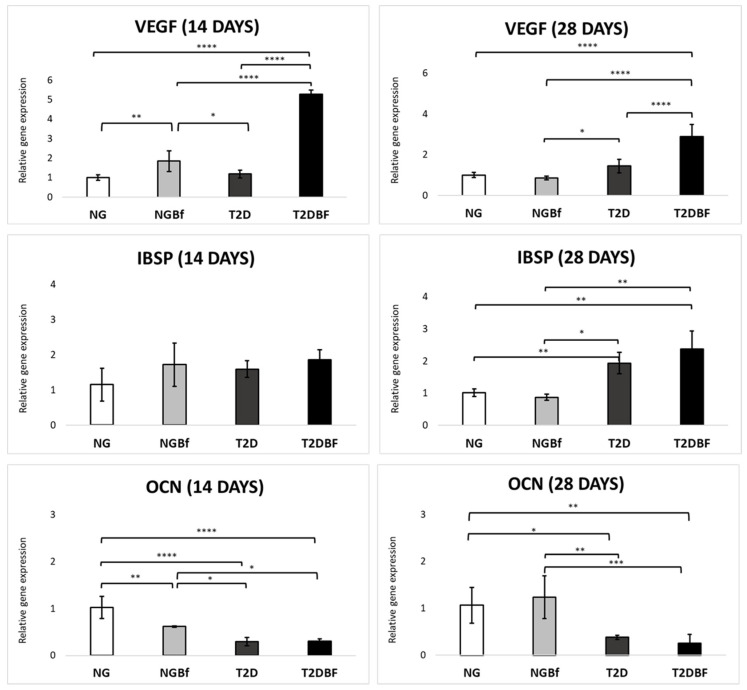
Molecular analysis results for the NG, NGBf, T2D, and T2DBf groups at 14 and 28 days of peri-implant repair. The *x*-axis shows the groups evaluated in this study. The *y*-axis shows the values of relative gene expression obtained on RT-qPCR analysis. The results of the one-way ANOVA statistical test (*p* < 0.05) are marked with brackets, where the higher the number of asterisks (*, **, ***, or ****), the higher the difference between the groups.

**Table 1 pharmaceuticals-18-01724-t001:** Average values of body weight for the NG, NGBf, T2D, and T2DBf groups.

	Day 0	Day 21	Day 28	Day 42	Day 70
**NG**	352.1 ± 31.05 ^a^	391.6 ± 137.3 ^ab^	364 ± 25 ^a^	365.5 ± 36.86 ^a^	459.5 ± 19 ^b^
**NGBf**	355.8 ± 28.09 ^a^	391.9 ± 137.3 ^a^	430.9 ± 29.46 ^a^	471.7 ± 29.45 ^b^	436.7 ± 139.3 ^a^
**T2D**	337.9 ± 17.67 ^a^	391.7 ± 38.57 ^a^	381.1 ± 29.35 ^a^	346.7 ± 35.55 ^a^	345.5 ± 36.18 ^a^
**T2DBf**	391.8 ± 42.99 ^a^	414.9 ± 44.41 ^a^	395.1 ± 30.26 ^a^	396.5 ± 72.12 ^a^	401.1 ± 93.09 ^a^

Body weight average of animals throughout the experiment. Letters correspond to weight comparison throughout the experiment. Data were analyzed using one-way ANOVA (*p* < 0.05). Values in grams.

**Table 2 pharmaceuticals-18-01724-t002:** Average values of blood glucose for the NG, NGBf, T2D, and T2DBf groups.

	Day 0	Day 21	Day 28	Day 42	Day 70
**NG**	94.25 ± 7.375 ^a^	94.75 ± 5.817 ^a^	97.17 ± 6.887 ^a^	99.33 ± 4.141 ^a^	98.33 ± 6.25 ^a^
**NGBf**	95.22 ± 7.092 ^a^	95.28 ± 5.592 ^a^	99.44 ± 5.853 ^a^	100.8 ± 6.119 ^a^	102.5 ± 4.905 ^b^
**T2D**	101.6 ± 4.997 ^a^	141.7 ± 22.04 ^b^	422.3 ± 48.48 ^ab^	395.9 ± 55.67 ^ab^	437 ± 6.33 ^ab^
**T2DBf**	102.1 ± 7.605 ^a^	126.2 ± 14.83 ^ac^	397.5 ± 75.79 ^b^	246.8 ± 113.3 ^cd^	185.1 ± 120.1 ^c^

Glycemic average of animals throughout the experiment. Letters correspond to weight comparisons throughout the experiment. One-way ANOVA statistical test (*p* < 0.05). Values in mg/dL.

**Table 3 pharmaceuticals-18-01724-t003:** Statistical differences presented in the confocal laser microscopy analysis in the NG, NGBf, T2D, and T2DBf groups.

	Comparison	*p* Value ^†^
Calcein	NG vs. DM2	0.0120
	NGBf vs. DM2	0.0257
Alizarin Red	NG vs. DM2	<0.0001
	NG vs. DM2Bf	0.0004
	NGBf vs. DM2	<0.0001
	NGBf vs. DM2Bf	0.0090

One-way ANOVA statistical test (*p* < 0.05). ^†^ Indicate statistically significant differences.

**Table 4 pharmaceuticals-18-01724-t004:** Nutritional information provided by *Bauhinia forficata* tea.

*Bauhinia forficata* (5 g)		
	Amount	Percentage (%)
**kcal**	0 kcal	0
**Carbohydrates**	0 g	0
**Proteins**	0 g	0
**Total Fat**	0 g	0
**Saturated Fat**	0 g	0
**Trans Fat**	0 g	(*)
**Dietary Fiber**	0 g	0
**Sodium**	0 g	0

Daily reference values based on a 2000 kcal diet. (*) % not established.

## Data Availability

The original contributions presented in this study are included in the article. Further inquiries can be directed to the corresponding authors.

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
