# Peer review of "Protective Effects of Bauhinia forficata on Bone Biomechanics in a Type 2 Diabetes Model"

_pharmaceuticals, 2025, doi:10.3390/ph18111724_

Round 1

Reviewer 1 Report

Comments and Suggestions for Authors

This study evaluated peri-implant and long bone biomechanics in type 2 diabetic animals treated with Bauhinia forficata. The authors induced type 2 diabetes in rats using a cafeteria diet combined with streptozotocin. Bauhinia forficata tea (50 g/L) was administered to the treatment groups, and various assays were performed to assess its therapeutic potential. The findings suggest that type 2 diabetes mellitus significantly compromises peri-implant bone repair, without affecting long bone metabolism, and that Bauhinia forficata exerts beneficial effects on both disease progression and tissue repair.

While the study contributes valuable data, several areas of the manuscript would benefit from clarification and revision:

  1. Lines 461–469: Please clarify whether the tea was provided to each animal individually or collectively to each group. How did the authors ensure that each rat consumed the same amount? If intake was variable, this could influence the results.
  2. Lines 457–460: The sentence in this section is unclear and should be rewritten for clarity.
  3. The body weight data would be more effectively presented as a line chart over time to clearly demonstrate trends.
  4. The manuscript states n = 8 per group. Please specify how many rats were sacrificed for each experiment (RT-qPCR, body weight monitoring, blood glucose measurements), as this is not entirely clear.
  5. While the study includes sham and treated groups, no untreated negative control group was used. In addition, was there a reason no positive control (e.g., a standard antidiabetic drug or an agent known to influence bone biomechanics in T2D models) was included? This would strengthen the comparative value of the findings.

Author Response

Initially, we appreciate the comments and points that the reviewer has made to our manuscript. The alterations in the text are highlighted in YELLOW.

This study evaluated peri-implant and long bone biomechanics in type 2 diabetic animals treated with Bauhinia forficata. The authors induced type 2 diabetes in rats using a cafeteria diet combined with streptozotocin. Bauhinia forficata tea (50 g/L) was administered to the treatment groups, and various assays were performed to assess its therapeutic potential. The findings suggest that type 2 diabetes mellitus significantly compromises peri-implant bone repair, without affecting long bone metabolism, and that Bauhinia forficata exerts beneficial effects on both disease progression and tissue repair.

While the study contributes valuable data, several areas of the manuscript would benefit from clarification and revision:

1 - Lines 461–469: Please clarify whether the tea was provided to each animal individually or collectively to each group. How did the authors ensure that each rat consumed the same amount? If intake was variable, this could influence the results.

RESPONSE: We believe that all animals consumed the available tea based on the blood glucose results presented in the study. Furthermore, every day, we found the bottles empty when we refilled them. We understand that this is a limitation of the study; however, the results presented were optimistic and indicate that all animals used the medication, either in greater or lesser amounts.

2 - Lines 457–460: The sentence in this section is unclear and should be rewritten for clarity.

RESPONSE: The sentence was modified for improved clarity during the text reading

3 - The body weight data would be more effectively presented as a line chart over time to clearly demonstrate trends.

RESPONSE: We would like to thank the reviewer for this suggestion. The graphs have been inserted into the text to facilitate interpretation of the results (Figures 1 and 2)

4 - The manuscript states n = 8 per group. Please specify how many rats were sacrificed for each experiment (RT-qPCR, body weight monitoring, blood glucose measurements), as this is not entirely clear.

RESPONSE: The same animals were used for all analysis. Blood samples were collected, and weights were measured weekly until the date of euthanasia, therefore the animals were not sacrificed to obtain these two results. For the other analyses, eight animals were used, whose right tibias were used to evaluate removal torque, and after implant removal, the peri-implant bone was collected for PCR. Microtomographic and confocal microscopy analysis were performed on the left tibia. In addition, the right femurs were collected for EMIC, while the left femurs were sent for microtomographic analysis. These facts were better explained in the text. (4.6 Euthanasia and Proposed Analyses)

5 - While the study includes sham and treated groups, no untreated negative control group was used. In addition, was there a reason no positive control (e.g., a standard antidiabetic drug or an agent known to influence bone biomechanics in T2D models) was included? This would strengthen the comparative value of the findings.

RESPONSE: The normoglycemic group acts as a negative control, since it does not receive any type of drug treatment. Therefore, we did not include an additional “untreated” group, thus avoiding duplication of controls and unnecessary use of animals. Furthermore, we chose to not include a pharmacological antidiabetic agent as a positive control because the primary objective of the study was to characterize the efficacy and biomechanical effects of the herbal medicine in isolation. The inclusion of medication at this point would greatly increase the number of animals. However, we recognize the importance of comparisons with pharmacological standards and intend to include a positive control in subsequent experiments to assess the comparative magnitude of the observed effects.

Reviewer 2 Report

Comments and Suggestions for Authors

  1. In the numerical descriptions of Table 1, there should be a decimal point instead of a comma.
  2. “Table 1. Average values of blood glucose for the NG, NGBf, T2D, and T2DBf groups”. The Table 1 must modify to Table 2.
  3. In the numerical descriptions of Table 2, there should be a decimal point instead of a comma.
  4. The x and y axes of Figure 1 are not clear and the axis explanations are not clear.
  5. The x and y axes of Figure 2 are not clear and the axis explanations are not clear.
  6. The x and y axes of Figure 5 are not clear and the axis explanations are not clear.
  7. The x and y axes of Figure 10 are not clear and the axis explanations are not clear.

Comments on the Quality of English Language

The English must be improved.

Author Response

REVIEWER 2

Initially, we appreciate the comments and points that the reviewer has made to our manuscript. The alterations in the text are highlighted in GREEN.

1 - In the numerical descriptions of Table 1, there should be a decimal point instead of a comma.

RESPONSE: We would like to thank the reviewer for observing the problem on the table. The modifications have been carried out in Table 1.

2 - “Table 1. Average values of blood glucose for the NG, NGBf, T2D, and T2DBf groups”. The Table 1 must modify to Table 2.

RESPONSE: We would like to thank the reviewer for observing the problem in the text. The modifications have been carried out.

3 - In the numerical descriptions of Table 2, there should be a decimal point instead of a comma.

RESPONSE: We would like to thank the reviewer for observing the problem in the table. The modifications have been carried out in Table 2.

4 - The x and y axes of Figure 1 are not clear and the axis explanations are not clear.

RESPONSE: The caption of figure 1 (Now is figure 3, due to the addition of two graphs– body weight and glycemic level) has been modified to improve understanding and interpretation of the results presented in the graphs.

5 - The x and y axes of Figure 2 are not clear and the axis explanations are not clear.

RESPONSE: The caption of figure 2 (Now is figure 4, due to the addition of two graphs– body weight and glycemic level) has been modified to improve understanding and interpretation of the results presented in the graphs.

6 - The x and y axes of Figure 5 are not clear and the axis explanations are not clear.

RESPONSE: The caption of figure 5 and the graphs has been modified to improve understanding and interpretation of the results presented in the graphs.

7 - The x and y axes of Figure 10 are not clear and the axis explanations are not clear.

RESPONSE: The caption of figure 10 (Now is figure 12, due to the addition of two graphs– body weight and glycemic level) has been modified to improve understanding and interpretation of the results presented in the graphs.

Reviewer 3 Report

Comments and Suggestions for Authors

The manuscript is well-structured and presents novel data supporting the therapeutic use of Bauhinia forficata in diabetic patients undergoing implant procedures. However, major revision is needed to consider publication

  1. The study's primary limitation is the lack of direct phytochemical analysis of the specific Bauhinia forficata tea batch used. While a detailed literature review is provided the composition and concentration of active compounds like kaempferitrin in the actual infusion administered to the rats remain unknown.
  2. The interpretation of the RT-qPCR results is somewhat superficial and, in parts, contradictory.
    1. VEGF: The T2DBf group shows the highest VEGF expression, which is interpreted positively as "greater angiogenic induction." However, persistently high VEGF at a later stage (28 days) could also indicate a prolonged or dysregulated inflammatory/repair phase. This dual role of VEGF should be discussed.
    2. OCN: The results show that OCN expression is highest in the NG and NGBf groups at 14 days but is significantly suppressed in the T2DBf group at 28 days. The authors state that proteins "may already be expressed" in diabetic animals, but this seems speculative and conflicts with the gene expression data.
  3. The central finding that T2D did not impair uninjured long bone metabolism (femur) is interesting but potentially under-discussed. The abstract and conclusions state this clearly, but the results show a non-significant trend towards worse biomechanical properties (Max Force, Stress) in the T2D and T2DBf groups compared to NGBf.
  4. The study concludes that the systemic condition of T2D did not alter the structural characteristics of uninjured long bones (femurs). However, the three-point bending test data reveal significant differences in mechanical properties. Please explain
  5. Add a dedicated paragraph in the discussion on the clinical implications and future research needs. Key questions to address: What would be the proposed equivalent human dosage? Should Bf tea be considered a primary treatment or an adjunct to conventional anti-diabetic drugs and improved glycemic control? How might it be integrated into the pre- and post-operative care of diabetic patients receiving implants?
  6. The discussion is lengthy and occasionally repetitive. Consider streamlining it by focusing on the main mechanistic pathways: 1) Glycemic Control, 2) Antioxidant/Anti-inflammatory effects, and 3) Direct bone anabolic effects (via OPG/RANKL, osteoblast gene expression).
  7. The sentence on page 11, "This indicates that Bauhinia forficata was effective in maintaining the weight of diabetic animals," could be clarified. The data in Table 1 shows T2DBf started heavier and maintained weight, while T2D lost weight post-STZ. It might be more accurate to say Bf prevented the weight loss typically associated with STZ-induced beta-cell toxicity.
  8. Section 4.4: It states "each animal received one implant in each tibia, totaling 72 implants." With 32 animals, this should be 64 implants. Please verify and correct this number
  9. Section 4.6.5 (RT-qPCR):The methods mention primers for VEGF, CD31, OCN, and BSP, but the results only present VEGF, IBSP (BSP), and OCN. Where are the CD31 results? Either present the data or remove CD31 from the methods to avoid confusion.

Author Response

Initially, we appreciate the reviewer’s insightful comments and constructive suggestions. The modificatins are highlighted in BLUE within the revised version

1 - The study's primary limitation is the lack of direct phytochemical analysis of the specific Bauhinia forficata tea batch used. While a detailed literature review is provided the composition and concentration of active compounds like kaempferitrin in the actual infusion administered to the rats remain unknown.

RESPONSE: We acknowledge the reviewer’s concern regarding the absence of direct phytochemical characterization of specific batch of Bauhinia forficata used in this study. Our experimental design aimed to replicate the traditional and translational use of this herbal medicine, the tea consumed by the population, rather than testing an isolated or standardized extract. For this reason, we prioritized biological and preclinical outcomes while referencing validated chromatographic and spectroscopic studies that have established the presence of kaempferitrin, quercetin, rutin and many other flavonoids as major bioactive constituents on Bauhinia forficata infusions [49-53].

Nevertheless, we recognize that batch-to-batch variability may occur and have now included a paragraph in the discussion acknowledging this as a methodical limitation and direction for future research, in which quantifications of phenolic compounds and kaempferitrin content will be incorporate o strengthen the translational reliability of findings.

2 - The interpretation of the RT-qPCR results is somewhat superficial and, in parts, contradictory.

A - VEGF: The T2DBf group shows the highest VEGF expression, which is interpreted positively as "greater angiogenic induction." However, persistently high VEGF at a later stage (28 days) could also indicate a prolonged or dysregulated inflammatory/repair phase. This dual role of VEGF should be discussed.

RESPONSE: We thank the reviewer for this important observation. We agree that VEGF plays a dual role during the bone repair process, promoting angiogenesis and osteogenesis in early phases, but potentially reflecting prolonged inflammation or delayed remodeling when persistently elevated at later stages. However, when considering the overall findings, particularly the concomitant improvements in bone microarchitecture (as observed in microtomography analysis) and mineral apposition (as observed in confocal analysis) in the group T2DBf, we interpret that the VEGF expression in this group, at both 14 and 28 days is an indicator of sustained angiogenic activity rather than a dysregulated inflammatory response

B - OCN: The results show that OCN expression is highest in the NG and NGBf groups at 14 days but is significantly suppressed in the T2DBf group at 28 days. The authors state that proteins "may already be expressed" in diabetic animals, but this seems speculative and conflicts with the gene expression data.

RESPONSE: We appreciate the suggestion and have streamlined the discussion.

3 - The central finding that T2D did not impair uninjured long bone metabolism (femur) is interesting but potentially under-discussed. The abstract and conclusions state this clearly, but the results show a non-significant trend towards worse biomechanical properties (Max Force, Stress) in the T2D and T2DBf groups compared to NGBf.

RESPONSE: We appreciate this important comment. The discussion has been expanded to clarify this part. The absence of statistically significant differences among the groups in the three point bending parameters indicates that the herbal medicine treatment did not negatively affect the mechanical integrity of uninjured long bones. This suggests that in the absence of surgical injury, type 2 diabetes did not significantly compromise the biomechanical of the femur. In contrast, when analyzing the microtomographic and biomechanical data from tibiae, a clear deterioration in microarchitectural bone tissue was observed in diabetic animals, particularly those untreated with Bauhinia forficata.

4 - The study concludes that the systemic condition of T2D did not alter the structural characteristics of uninjured long bones (femurs). However, the three-point bending test data reveal significant differences in mechanical properties. Please explain

RESPONSE: We appreciate the reviewer’s comment. Some statistically significant variations in the three-point bending parameters were observed but remained within a small magnitude. These differences likely reflect biological variability rather than structural impairment.

5 - Add a dedicated paragraph in the discussion on the clinical implications and future research needs. Key questions to address: What would be the proposed equivalent human dosage? Should Bf tea be considered a primary treatment or an adjunct to conventional anti-diabetic drugs and improved glycemic control? How might it be integrated into the pre- and post-operative care of diabetic patients receiving implants?

RESPONSE: In accordance with the reviewer, a new paragraph has been added to the Discussion.

6 - The discussion is lengthy and occasionally repetitive. Consider streamlining it by focusing on the main mechanistic pathways: 1) Glycemic Control, 2) Antioxidant/Anti-inflammatory effects, and 3) Direct bone anabolic effects (via OPG/RANKL, osteoblast gene expression).

RESPONSE: We appreciate the suggestion and have streamlined the discussion.

7 - The sentence on page 11, "This indicates that Bauhinia forficata was effective in maintaining the weight of diabetic animals," could be clarified. The data in Table 1 shows T2DBf started heavier and maintained weight, while T2D lost weight post-STZ. It might be more accurate to say Bf prevented the weight loss typically associated with STZ-induced beta-cell toxicity.

RESPONSE: We thank the reviewer for identifying this point. The sentence has been revised for greater accuracy ad follows: “This suggests that Bauhinia forficata prevented the weight loss typically associated with STZ-induced -cell toxicity, helping maintain weight stability in diabetic animals”. The authors agreed that this change provides a more precise reflection of the data shown in Tabele 1

8 - Section 4.4: It states "each animal received one implant in each tibia, totaling 72 implants." With 32 animals, this should be 64 implants. Please verify and correct this number

RESPONSE: We appreciate the reviewer’s attention to this numerical inconsistency. The number of implants has been corrected to 64 in section 4.4. (implant installation) to accuratelu represent the two implants placed per animal (32 x 2 =64)

9 - Section 4.6.5 (RT-qPCR): The methods mention primers for VEGF, CD31, OCN, and BSP, but the results only present VEGF, IBSP (BSP), and OCN. Where are the CD31 results? Either present the data or remove CD31 from the methods to avoid confusion.

RESPONSE: We agree that the inclusion of CD31 in the Methods without corresponding data could lead to confusion. Since the CD31 gene expression results were not part of the final dataset due to low amplification efficiency, we have removed this marker from the methods section to maintain methodological consistency.

Reviewer 4 Report

Comments and Suggestions for Authors

The current research article represents a new approach to herbal medicine and investigates its effect on the bone biomechanics in a type 2 diabetes model. However, it needs major revision to be suitable for publication.

In the abstract: Include values of outstanding results coming out of the utilisation of Bauhinia forficata.

In the Introduction:

  • Provide a more detailed explanation of the available treatment approaches.

  • The link between the problem (study focus) and herbal medicine is missing.

  • Include different examples of herbal medicine for the treatment of bone fractures.

in the results section:

  • Please enhance the resolutions of Figures 2 and 5 and expand them to fill the entire page.

  • Add some notifications to Figures 3, 6, and 7 to make them more readable for readers.

  • Change the figure title from capital letters to normal in figures 8 and 9. Also included the meaning of letters A and B in their captions.

  • Explain the meaning of (*and ***) in the caption of Figure 10.

In the methodology section:

  • Include sample preparation and measurement method in section ''4.3.1''.

  • The references should be within the paragraphs, not after the end of the paragraphs. Please check this in the whole manuscript.

The conclusions

State in the conclusion the negative effect of BF on the type 2 diabetic rat treatments.

Author Response

Initially, we appreciate the comments and points that the reviewer has made to our manuscript. The alterations in the text are highlighted in PINK.

The current research article represents a new approach to herbal medicine and investigates its effect on the bone biomechanics in a type 2 diabetes model. However, it needs major revision to be suitable for publication.

1 - In the abstract: Include values of outstanding results coming out of the utilization of Bauhinia forficata.

RESPONSE: We thank the reviewer for this valuable suggestion. We agree that including quantitative data in the abstract could enhance the clarity of the main findings. However, due to the strict word/character limit imposed by the journal and the extensive number of outcomes assessed, we prioritized maintaining a concise and readable abstract that highlights the principal trends and conclusions. All corresponding numerical values are fully presented in the Results section, allowing readers to access the detailed data while keeping the abstract within the journal’s length constraints. If necessary and the journal accept the extensive abstract, we can include one representative quantitative result in the abstract to illustrate the magnitude of the observed effects.

2 - In the Introduction:

A - Provide a more detailed explanation of the available treatment approaches.

RESPONSE: We are grateful for the suggestion made by the reviewer to include the available treatment approaches for type 2 diabetes. New information on conventional and phytotherapeutic treatments has been included in the introduction. (Lines 68-80).

B - The link between the problem (study focus) and herbal medicine is missing.

RESPONSE: More information about the importance of herbal medicine and its effect on type 2 diabetes and bone was added to the text, providing a link to the objective of this study.

C - Include different examples of herbal medicine for the treatment of bone fractures.

RESPONSE: Further studies with natural compounds and their activity on bone metabolism were added to the text. (lines 92-104)

3 - In the results section:

A - Please enhance the resolutions of Figures 2 and 5 and expand them to fill the entire page.

RESPONSE: The two figures have been reformulated to improve visualization in the text.

B - Add some notifications to Figures 3, 6, and 7 to make them more readable for readers.

RESPONSE: More information has been added to the figures mentioned by the reviewer.

C - Change the figure title from capital letters to normal in figures 8 and 9. Also included the meaning of letters A and B in their captions.

RESPONSE: The necessary explanations are provided in Figures 8 and 9.

D - Explain the meaning of (*and ***) in the caption of Figure 10.

RESPONSE: This information was added to the figures as requested by another reviewer (highlighted in green in the captions).

4 - In the methodology section:

A - Include sample preparation and measurement method in section ''4.3.1''.

RESPONSE: We appreciate the reviewer’s observation regarding the preparation and analytical methods of Bauhinia forficata. In the present study, our main objective was to reproduce the traditional form of use of this plant, as consumed by the population, and evaluate its biological effects in an experimental diabetic model, as mentioned in the section 4.3.1. For this reason, no new chromatographic or spectroscopic analyses were performed. Instead, the chemical profile was established based on well-documented studies in the literature [49–53], which consistently describe the presence of flavonoids (such as kaempferitrin, quercetin, and rutin), lactones, terpenoids, and glycolipids in both the leaves and aqueous infusions of B. forficata. Since these compounds have been repeatedly reported and quantified in previous validated analytical protocols, we used this bibliographic evidence to ensure translational relevance and reproducibility of the preparation, maintaining our focus on the biological and preclinical outcomes rather than on re-characterization of the extract composition. Additional information about the mentioned has been added to the text. (lines 564-568)

B - The references should be within the paragraphs, not after the end of the paragraphs. Please check this in the whole manuscript

RESPONSE: We are grateful to the reviewer for this information. The references have been placed according to their appearance in the text and in accordance with the manuscript guidelines.

5 - The conclusions:

A - State in the conclusion the negative effect of BF on the type 2 diabetic rat treatments.

RESPONSE: Once again, we thank the reviewer for the suggestion. No collateral effects were observed in normoglycemic animals treated with Bauhinia, and this fact was added to the conclusion.

Round 2

Reviewer 2 Report

Comments and Suggestions for Authors

The authors had reply the questions of reviewer.

Comments on the Quality of English Language

The Enhlish writing ca be improved.

Author Response

Initially, we appreciate the comments and points that the reviewer has made to our manuscript. The alterations in the text are highlighted in DARK BLUE.

1 - Comments and Suggestions for Authors: The authors had reply the questions of reviewer.
2 - Comments on the Quality of English Language: The Enhlish writing ca be improved.

RESPONSE: We sincerely thank the reviewer for their positive evaluation. In response to the suggestion, the entire manuscript has undergone a comprehensive English language revision to improve it. The content was carefully reviewed

Reviewer 4 Report

Comments and Suggestions for Authors

The authors have covered most of the raised issues, but only the included figure does not contain standard deviation or error bars. When they include it, the paper will be accepted.

Author Response

Comments and Suggestions for Authors: The authors have covered most of the raised issues, but only the included figure does not contain standard deviation or error bars. When they include it, the paper will be accepted.

RESPONSE: We thank the reviewer for the attentive observation. We would like to clarify that all graphs already included error bars representing the standard deviation. In addition, for the analyses of body weight and blood glucose levels, we represented the mean values in the graphs, while the standard deviations are displayed in the corresponding tables (Tables 1 and 2). This approach was chosen to improve data visualization and to avoid overlapping information between graphs and tables.